# Soft Computing Approach for Predicting the Effects of Waste Rubber–Bitumen Interaction Phenomena on the Viscosity of Rubberized Bitumen

Michele Lanotte 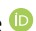

Department of Civil Infrastructure and Environmental Engineering, Khalifa University of Science and Technology, Abu Dhabi 127788, United Arab Emirates; michele.lanotte@ku.ac.ae

**Abstract:** The ability to anticipate the effects of the interaction between waste rubber particles from end-of-life tires and bitumen can encourage the use of rubberized bitumen, a material with proven environmental benefits, in civil engineering applications. In this study, a predictive model of rubberized bitumen viscosity is presented for this purpose. A machine learning-based approach (Multi-Gene Genetic Programming—MGGP) and a more traditional multi-variable least square regression (MLSR) method are compared. The statistical analysis indicates that the robustness and the capability of the MGGP algorithm led to a better estimation of the rubberized bitumen's viscosity. Additionally, the MGGP analysis returned an actual equation that could be easily implemented in any spreadsheet for an initial tuning of the production protocol based on the desired level of interaction between the rubber and bitumen.

**Keywords:** waste tire rubber; rubberized asphalt; viscosity; manufacturing optimization; multi-gene genetic programming

## 1. Introduction

The use of waste tire rubber in bitumen for road paving applications has been evaluated extensively by several research groups around the world from both mechanical performance [1–7] and environmental perspectives [8,9]. Since the discovery of this technology in the early 1960s, the use of crumb rubber (CR) as a bitumen modifier has experienced a significant evolution. However, its field implementation is still limited compared to the claimed benefits. One of the reasons is that the interaction phenomena between rubber and bitumen are highly variable and the control of the modified bitumen production is troublesome.

In general terms, the interaction starts as soon as the CR particles come into contact with the asphalt bitumen at high temperatures (150–200 °C). Due to the diffusion of the oily fractions of the bitumen into the rubber network, a layer of gel-like material forms in the outer part of the particles. The extension of this layer extends from the rubber–bitumen contact surface to the maximum point of penetration of the oily fraction into the rubber and leads to the swelling of the particles [10]. The presence of the rubber cross-linked chains prevents the degradation of the particles for a limited time only. Then, the high temperature and mechanical stirring action initiate the devulcanization and depolymerization process of the rubber. Some components (e.g., carbon black) are now released into the bitumen matrix and the fragmentation of the rubber particles starts [10–13]. The magnitude of the above-mentioned phenomena is extremely variable and the production protocol plays a fundamental role. Rubber particles, in fact, react in a time-temperature dependent manner [14]. Swelling and degradation can speed up or slow down by either increasing or decreasing the blending time and temperature [15,16]. The physical and chemical characteristics of the rubber and the chemical composition of the base binder also play a crucial role in the interaction. In fact, the higher the surface area, the higher the quantity of

oil that can be absorbed by the rubber, if all other variables are kept constant. Then, the maximum degree of swelling can be reached in a shorter time [14].

According to the IBISWorld Industry Report on Tire and Rubber Recycling released in September 2018, in the US only, there are more than 100 crumb-rubber production facilities [17]. A study conducted in Italy by Ecopneus, reported 20 manufacturers selling crumb rubber in a territory that is a hundred times smaller than the US [18]. Each of these factories produces more than one type of crumb rubber, which leads to many different CR available on the market. When the purchase decision in merely economic, the same bitumen refinery may use different types of crumb rubber in the same paving season.

Regardless of the type of materials used, all interaction processes have something in common. In the first stage, the simultaneous reduction in the inter-particle distance due to the swelling action leads to an overall increase in viscosity [11], whereas in the second stage, as rubber starts to degrade, the particles become smaller with a consequent reduction in viscosity. Figure 1 shows the mixing phases and their effect on the viscosity of the modified bitumen. As such, viscosity, even though indirectly, can be an indicator of the interaction phenomena.

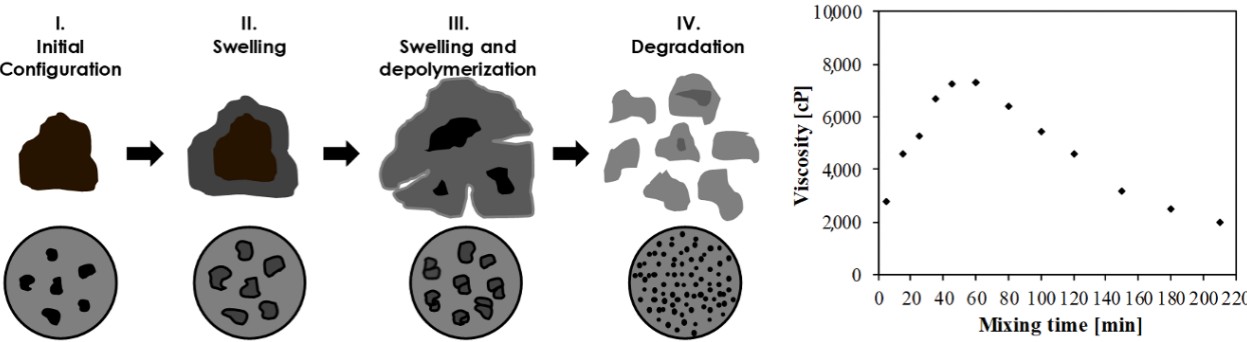

**Figure 1.** Particle size distribution of the CRs employed for laboratory preparation of AR binders.

Where an asphalt plant is well equipped, material engineers can monitor the status of the modification by measuring the viscosity of the binder during the blending process and stop it when the desired degree of interaction has been met. However, the correct degree of interaction must be known in advance since the same viscosity can be obtained in the swelling and degradation phases (Figure 1) but the mechanical characteristics are substantially different [14]. The ability to anticipate the result of the blending process for a given combination of binder and crumb rubber and a given set of processing conditions can represent a useful tool for practitioners. This research contributes to the ongoing efforts on this subject [19–21] and presents a mathematical equation for predicting viscosity as a function of mixing time, mixing temperature, and crumb-rubber characteristics.

Machine learning methodologies are extensively used to develop predictive algorithms for a material's characteristics [19–21]. Despite their significant contribution, the algorithms generated by some of these methodologies are not easily implementable (e.g., neural networks, fuzzy logic algorithms) since they necessitate a knowledge of programming. For this reason, the model presented in this paper is developed by a machine learning approach named Multi-Gene Genetic Programming (MGGP), which returns an actual equation that can be easily implemented in any worksheet [22–24].

## 2. Principle of Genetic Programming

In 1992, Koza introduced the genetic programming (GP) approach as a branch of genetic algorithms (GA) [25]. GP is a computation technique that creates and evolves computer programs by implementing the "Darwinian natural selection" principles to obtain the best solution for a given problem. The algorithm randomly generates a primary population of computer programs (e.g., mathematical equations) structured like trees, where function nodes (e.g., arithmetic operations, Boolean logic functions, mathematical

functions) are used to connect terminal nodes (variables and numerical constants). The goodness-of-fit of all the individuals in the computer programs' population is evaluated, and the ones with superior performance are selected and carried to the next generation. The next generation is then created starting from those individuals and evolves by genetic operations such as mutation, crossover, and reproduction. Since this is a randomly based process, some of the individuals may move to the next generation without any evolution during reproduction, whereas others can evolve extensively [26].

Multi-Gene Genetic programming (MGGP) is an extension of the GP approach. The above-mentioned trees, also known as genes, which usually contain nonlinear terms, are here combined linearly with each other using the least square method. This helps to incorporate the power of both linear and nonlinear regression approaches to enhance the performance of the algorithm. However, the number of genes and their depths must be controlled by the user to avoid a further increase in equation complexity and decrease in algorithm efficiency.

In this research, the MGGP algorithm implemented in the GPTIPS software was used [27] as it allows changing some parameters of the algorithm manually to find an acceptable balance between the running time, output complexity, and goodness-of-fit of the models. The software settings were initially chosen based on recommendations provided in the literature [23,24] and then adapted to the database of materials' characteristics used in this study and described in the next paragraph. As indicated in Table 1, the initial population size is one of the main inputs of the analysis. This represents the number of equations initially developed by the software and carried over the analysis. The final output is the series of equations that reached the final evolutionary step, each of them characterized by its goodness-of-fit and expressional complexity. Therefore, the user can either decide to consider all the equations or analyze the equations in a certain range of these two values.

**Table 1.** Parameter settings for MGGP algorithm.

| Parameter | Settings |
|---|---|
| Function set | $+, -, \times, \div, sqr, power, Ln$ |
| Population size | 600 |
| Number of generations | 100 |
| Maximum number of genes allowed in an individual | 8 |
| Maximum tree depth | 8 |
| Tournament size | 25 |
| Elitism | 0.01% of population |
| Crossover events | 0.85 |
| High-level crossover | 0.2 |
| Low-level crossover | 0.8 |
| Mutation events | 0.1 |
| Sub-tree mutation | 0.9 |
| Replacing input terminal with another random terminal | 0.05 |
| Gaussian perturbation of randomly selected constant | 0.05 |
| Direct reproduction | 0.05 |
| Ephemeral random constants | [−10 10] |

## 3. Materials and Methods

Four crumb rubbers (CRs) sampled from three end-of-life tire (ELT) processing plants were characterized in a laboratory in terms of particle size distribution, density, and surface area. Two recycling plants used the so-called 'ambient' particle size reduction process, whereas the third one processed ELTs under cryogenic conditions. The nominal maximum size of the CR particles produced with the ambient process was 1 mm and 0.5 mm for the samples named CR1 and CR2, respectively. The cryogenic product, named CR4, had a nominal maximum size of 0.6 mm. CR1, CR2, and CR4 were used as collected in the plant. CR3, instead, was obtained in a laboratory as a sub-product of CR1 by eliminating



particles retained at a 0.354 mm sieve. This additional step was necessary to have information about the interaction effects due to CR with the same chemical composition but different gradations.

CR gradation was determined by a sieve analysis performed in dry conditions adopting the ASTM series sieve (1, 0.841, 0.71, 0.589, 0.5, 0.354, 0.25, 0.177, 0.125, 0.088, 0.063 mm). The results were then fitted to the Weibull distribution:

$$P_d = 1 - e^{-(d/\lambda)^k} \tag{1}$$

where $d$ is the sieve opening (in mm), $P_d$ is the percentage passing to the sieve through the opening $d$, and $\lambda$ and k are the Weibull distribution parameters. The fitting process allowed the use of $\lambda$ and k as the synthetic indicators of the shape curve distribution and uniformity of the particles' distribution, respectively.

The particles' density was measured using the pycnometer method, with ethyl alcohol employed as the reference fluid since its density is lower than that of rubber and therefore prevents CR particles from floating to the surface.

The assessment of the surface area (SA) per unit mass was based on digital images of the particles retained at each sieve size generated using a stereomicroscope. Pictures captured using a digital camera were subsequently processed through the software ImageJ. The following equation, proposed by Santagata et al. [15], was used for the SA estimation:

$$SA_m = \phi(6/\rho)\Sigma_i(f_1/d_{m,i}) \tag{2}$$

where $\phi$ is a correction factor, which is a function of the particles' shape and roughness, $\rho$ is the density (in g/m$^3$), $f_i$ is the frequency (in decimal units) of the $i$-th single-size fraction, $d_{m,i}$ is the mean particle diameter (in mm) of the i-th fraction.

CRM binders were prepared to combine each CR with a standard Pen 50/70 bitumen at three different dosages (15%, 18.5%, and 22% by weight of the base bitumen). Rubber particles stored at room temperature were added to the pre-heated base bitumen in batches immersed in a thermostatic oil bath in order to maintain a constant temperature of 190 °C. Blends were then stirred by an anchor-shaped spindle for a total time of 210 min at a speed of 600 RPM. During the mixing process, the CRM binders were sampled at predefined time intervals (5, 15, 25, 35, 45, 60, 80, 100, 120, 150, 180, 210 min) and immediately subjected to viscosity tests to assess the evolution of viscosity as a function of the mixing time. The same mixing, sampling, and testing procedures were used for the blends produced at 150 °C and 170 °C with a crumb-rubber dosage of 18.5% only.

Rotational viscosity was measured with a Brookfield viscometer DVIII-Ultra equipped with an SC4-27 spindle. The modified bitumen was sampled from the blending container, poured into the viscometer cup, and tested at a single temperature of 175 °C (as specified in the ASTM D-6114). Three subsequent steps of rotational speeds characterized by different durations were considered. In the first step, the rotational speed was set at 10 RPM (corresponding to a shear rate of 3.4 s$^{-1}$) for six minutes to allow the transition from mixing to testing temperature and reach the steady-state flow conditions. The second and third stages had a duration of one minute, and the rotational speed was increased to 20 RPM (6.8 s$^{-1}$) and 50 RPM (17 s$^{-1}$), respectively. Only data recorded at 20 RPM are considered in this study since they correspond to the shear rate indicated in the ASTM specifications. Other data were used for different purposes.

## 4. Development of the Predictive Model

The development of the predictive model has been divided into two phases. In Phase I, the results of a Multi-Variable Least Square Regression (MLSR) model were compared to a preliminary MGGP-based model to highlight the benefits of using the MGGP algorithm. In addition, the variables' inter-dependency and possible anomalies were analyzed. The MGGP algorithm was then used for a refined analysis in Phase II. The quality of the

prediction was assessed by calculating the coefficient of determination ($R^2$), mean squared error (MSE), and mean absolute error (MAE) as follows:

$$R^2 = 1 - \sum_{i=1}^{n} \frac{(h_i - t_i)^2}{\sum_{i=1}^{n}(h_i - \widehat{h})^2} \tag{3}$$

$$\text{RMSE} = \sqrt{\sum_{i=1}^{n} \frac{(h_i - t_i)^2}{n}} \tag{4}$$

$$\text{MAE} = \sum_{i=1}^{n} \frac{|h_i - t_i|^2}{n} \tag{5}$$

where $h_i$ is the experimental outputs, $t_i$ is the calculated outputs, $\hat{h}$ is the average of the experimental outputs, $t_i$ is the average of the calculated outputs, and n is the number of samples.

### 4.1. Phase I: Preliminary Analyses with MGGP and MLSR Approaches

The inter-dependency of the variables was considered to avoid overstating the effects on the mathematical model. An R-value of 0.8 between two variables was specified in previous studies as a good threshold [28] to discriminate variables' inter-dependency. The results indicated high correlations between three parameters: $\lambda$, k, and SA. Hence, a database should contain only one of them or none. Four distinct databases were created for the modeling process:

- Set 1: Weibull parameter $\lambda$, density ($\lambda$, g/cm$^3$), mixing temperature (T, °C), mixing time (t, min), and rubber content (RC, %),
- Set 2: Weibull parameter k, density, mixing temperature, mixing time, and rubber content,
- Set 3: Surface area (SA, mm$^2$), density, mixing temperature, mixing time, and rubber content,
- Set 4: Weibull parameter k, mixing temperature, mixing time, and rubber content.

The set of data used for the preliminary MLSR and MGGP analyses had a population of 222 viscosity test results that were randomly divided into two subsets: 80% of the population was utilized for training and the remaining 20% was considered for validating (testing) the models. Since the purpose of the study was not the description of these materials from a mechanical point of view, only the statistical description of the data collected and used for modeling is herein reported (Table 2).

**Table 2.** Descriptive statistics of the model variables for the entire database.

| Parameter | $\lambda$ | k | SA [mm$^2$] | $\rho$ [g/cm$^3$] | T [°C] | t [min] | RC [%] | $\eta$ [cP] |
|---|---|---|---|---|---|---|---|---|
| Mean | 0.409 | 3.46 | 225 | 1.201 | 177.2 | 86.1 | 18.5 | 4883 |
| Median | 0.402 | 2.17 | 221 | 1.196 | 190.0 | 70.0 | 18.5 | 4295 |
| Maximum | 0.580 | 6.21 | 386 | 1.223 | 190.0 | 210.0 | 22.0 | 15,715 |
| Minimum | 0.254 | 1.91 | 113 | 1.180 | 150.0 | 5.0 | 15.0 | 945 |
| Std.Dev. | 0.114 | 1.69 | 102 | 0.016 | 16.21 | 65.8 | 2.13 | 3057 |
| Skewness | 0.156 | 0.72 | 0.67 | 0.084 | −0.744 | 0.524 | 0.004 | 1.619 |
| Kurtosis | 2.054 | 1.97 | 1.97 | 1.808 | 1.928 | 1.984 | 2.707 | 5.632 |
| Sum | 90.7 | 767 | 49,951 | 266.7 | 39,340 | 19,110 | 4100 | $1.1 \times 10^6$ |
| ΣSq.Dev. | 2.894 | 627 | 2.30 | 0.055 | 58,068 | $9.5 \times 10^6$ | 1004 | $2.1 \times 10^9$ |
| Observations | 222 | 222 | 222 | 222 | 222 | 222 | 222 | 222 |

The MLSR and MGGP models that resulted from the preliminary analysis are

$$\eta = -112\lambda - 1600\rho + 37.6T - 3.9t + 1001.5RC - 17588.6 \tag{6}$$

$$\eta = 110T - 1688\lambda + 215t - 1688RC - 1577Ln(t^3) + 122RC \cdot Ln(t^3) - 0.913T \cdot t$$
$$- 0.913T \cdot RC - 0.913t \cdot RC^2 + 2.33RC^3 + 2.33\rho \cdot t \cdot RC - 490 \quad (7)$$

The best MLSR- and MGGP-based equations are both functions of the rubber content, mixing time, mixing temperature, rubber density, and Weibull parameter $\lambda$ (Set 1).

Figure 2 shows the relationship between the predicted and measured viscosity values using the MLSR and MGGP approaches. It is evident that the MLSR approach, given its mathematical form, was unable to develop an adequate equation to forecast the viscosity of the rubberized binders. In Figure 2a, data points are presented with three different symbols to highlight the gaps in the predictions for values between 2500 cP and 3200 cP as well as in the range 5800–8200 cP. On the other hand, the MGGP model shows a good overall goodness of fit. However, a difference existed for viscosity values above 6000 cP, which were mainly recorded for the binders with the highest crumb-rubber content. Thus, some of the variables considered in the analysis have a specific significance and impact the different mixing conditions, and there was a certain level of particle swelling and degradation. For this reason, the threshold value of 6000 cP was selected to split the initial database and refine the analysis in Phase II.

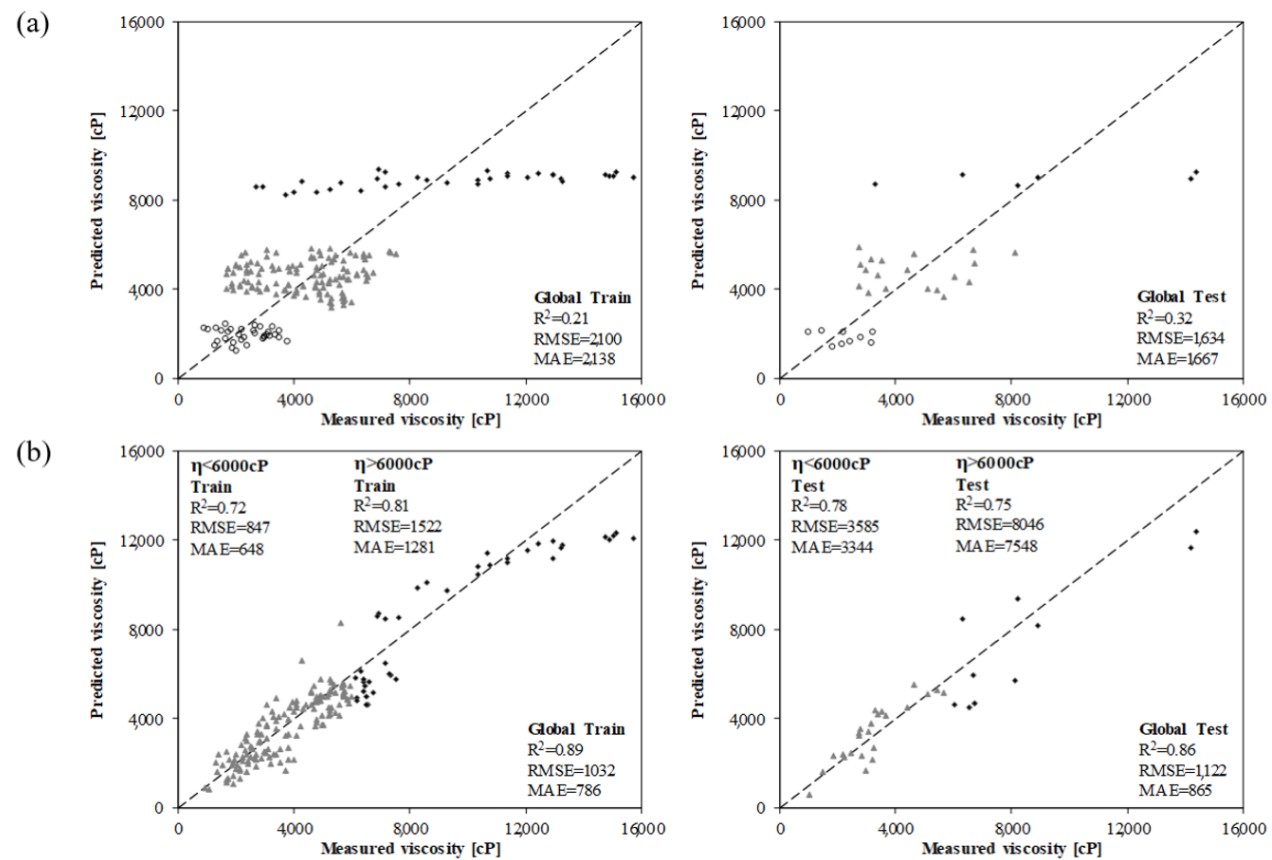

**Figure 2.** Preliminary results of binder viscosity prediction using the (**a**) MLSR and (**b**) MGGP models.

### 4.2. Phase II: Refined MGGP Model

In this second round of the modeling process, 24 values of the initial database were removed to be used for the final validation of the predictive models. The remaining 198 individual values were divided among those below (153 values) and above (45 values) 6000 cP. The statistical descriptions of these two sub-databases are provided in Tables 3 and 4. The percentages for the subdivisions of each database for training and testing purposes were kept as per the previous step of the analysis.

**Table 3.** Descriptive statistics of the model variables for $\eta < 6000$ cP.

| Parameter | $\lambda$ | k | SA [mm$^2$] | $\rho$ [g/cm$^3$] | T [°C] | t [min] | RC [%] | $\eta$ [cP] |
|---|---|---|---|---|---|---|---|---|
| Mean | 0.413 | 3.430 | 219.0 | 1.202 | 173.9 | 92.3 | 17.8 | 3544 |
| Median | 0.413 | 3.726 | 171.0 | 1.205 | 190.0 | 80.0 | 18.5 | 3300 |
| Maximum | 0.580 | 6.213 | 386.0 | 1.223 | 190.0 | 210.0 | 22.0 | 5975 |
| Minimum | 0.254 | 1.908 | 113.0 | 1.180 | 150.0 | 5.0 | 15.0 | 945 |
| Std.Dev. | 0.107 | 1.613 | 95.5 | 0.017 | 17.63 | 70.4 | 1.88 | 1372 |
| Skewness | 0.156 | 0.778 | 0.839 | 0.032 | −0.392 | 0.37 | −0.13 | 0.123 |
| Kurtosis | 2.359 | 2.165 | 2.387 | 1.628 | 1.408 | 1.68 | 2.84 | 1.825 |
| Sum | 68.82 | 524.2 | 33,502 | 183.8 | 26,610 | 14,120 | 2715 | $5.4 \times 10^5$ |
| ΣSq.Dev. | 1.730 | 395.3 | $1.4 \times 10^6$ | 0.042 | $4.7 \times 10^4$ | $7.5 \times 10^5$ | 537.5 | $2.9 \times 10^8$ |
| Observations | 153 | 153 | 153 | 153 | 153 | 153 | 153 | 153 |

**Table 4.** Descriptive statistics of the model variables for $\eta > 6000$ cP.

| Parameter | $\lambda$ | k | SA [mm$^2$] | $\rho$ [g/cm$^3$] | T [°C] | t [min] | RC [%] | $\eta$ [cP] |
|---|---|---|---|---|---|---|---|---|
| Mean | 0.393 | 3.230 | 232.5 | 1.200 | 186.9 | 65.3 | 20.9 | 9549 |
| Median | 0.402 | 2.167 | 221.0 | 1.196 | 190.0 | 60.0 | 22.0 | 8245 |
| Maximum | 0.580 | 6.213 | 386.0 | 1.223 | 190.0 | 150.0 | 22.0 | 15,715 |
| Minimum | 0.254 | 1.908 | 113.0 | 1.180 | 170.0 | 5.0 | 18.5 | 6025 |
| Std.Dev. | 0.108 | 1.580 | 100.5 | 0.016 | 7.331 | 41.8 | 1.64 | 3219 |
| Skewness | 0.250 | 0.988 | 0.612 | 0.184 | −1.90 | 0.34 | −0.82 | 0.533 |
| Kurtosis | 2.357 | 2.558 | 1.932 | 1.711 | 4.613 | 2.19 | 1.67 | 1.792 |
| Sum | 17.87 | 145.5 | 10,461 | 54.01 | 8410 | 2940 | 941 | $4.2 \times 10^5$ |
| ΣSq.Dev. | 1.730 | 395.3 | $1.4 \times 10^6$ | 0.042 | $4.7 \times 10^4$ | $7.5 \times 10^5$ | 537.5 | $2.9 \times 10^8$ |
| Observations | 45 | 45 | 45 | 45 | 45 | 45 | 45 | 45 |

As reported in Table 1, 600 equations were developed for every generation of the analysis. The frequency of occurrence of each variable was calculated (frequency equal to one indicates that all equations contain that variable). The results of the sensitivity analysis are illustrated in Figure 3. The mixing time and rubber content had the most dominant effect on the CRM binder viscosity prediction, followed by the mixing temperature and rubber density. The gradation also played an important role in forecasting the viscosity values. None of the equations for viscosity above 6000 cP considered the surface area. As mentioned in the previous section, high-viscosity values are usually recorded for a high rubber content, which may be a condition in which the surface area could lose its significance. The effect of a single parameter on viscosity can also be analyzed by a parametric study. Figure 4 illustrates the impact on the forecast viscosity due to the variation in the rubber density, Weibull parameter $\lambda$, and k. The parametric study of the surface area is not shown for the reason mentioned above. The effect of the gradation changed considerably in the two scenarios, whereas the rubber density had the same effect. However, the inherent randomness of the MGGP algorithm did not allow for the control of or force the effect of a given parameter in order to have the desired trend, and the algorithm optimized the prediction models as a result of the superposition of all the effects of the variables analyzed.

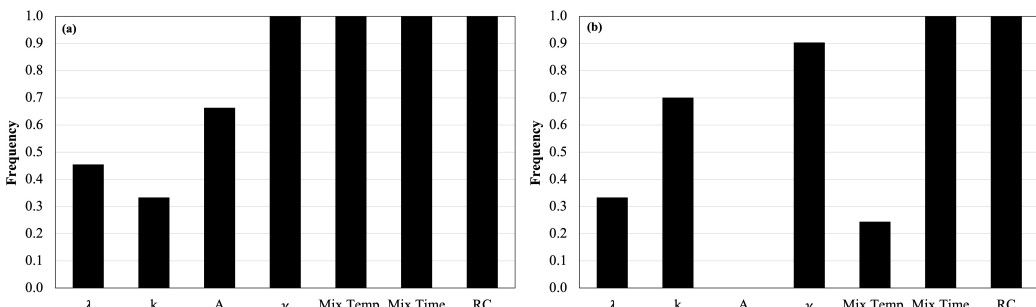

**Figure 3.** Frequency of input parameters: (**a**) $\eta < 6000$ cP, and (**b**) $\eta > 6000$ cP.

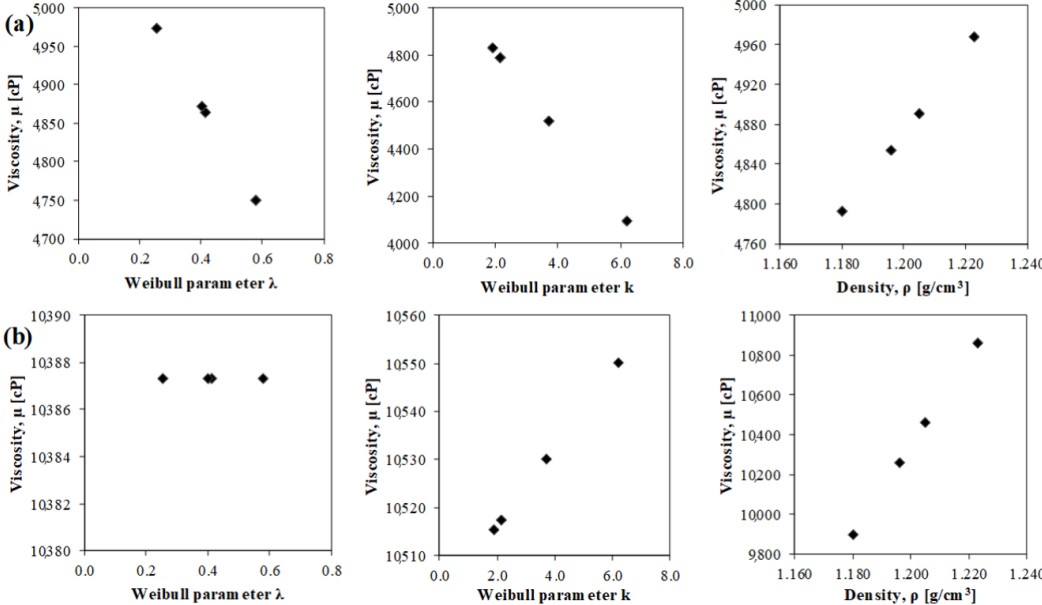

**Figure 4.** Parametric study: (**a**) $\eta < 6000$ cP, and (**b**) $\eta > 6000$ cP.

*4.3. Predictive Models and Performance Analysis*

For each data set listed previously, the best equation was selected based on the goodness of fit and the complexity of the analytical equation. The statistics of the MGGP equations are reported in Table 5. Based on the sum of the MAE for the training and test data, it can also be concluded that the models obtained with the third and fourth sets of inputs had the highest accuracy for predicting the CRM binder viscosity below and over the viscosity threshold. The final predictive models can be then expressed as in Equation (8). The development of two distinct MGGP predictive models for the CRM binder viscosity increased the overall goodness of fit as shown in Figure 5a,b.

$$
\eta = \begin{cases}
\begin{aligned}
& 469T - 11,900\rho + 20.3t + 186RC + 1944\sqrt{t + 2RC} \\
& -1.28T(t - 9.06\rho + Ln(k)) - 1.28T^2 + 0.524\rho^{1.5} \cdot t(2\rho + T) - 43,700 \\
& \qquad (for\ \eta < 6000\ \text{cP})
\end{aligned} \\[2em]
\begin{aligned}
& 8.08\left(k + T + RC^2\right) + 270t + 1900RC + 62,000 Ln(t) - 25,800\sqrt{t} \\
& -\frac{7377\sqrt{\frac{T}{RC}} + 7377RC + 53,800}{t} + \frac{1344t \cdot Ln(t)}{RC} - 1,230,000) \\
& \qquad (for\ \eta > 6000\ \text{cP})
\end{aligned}
\end{cases}
\tag{8}
$$

**Table 5.** Summary of the statistical performance of the MGGP models.

| | | Set 1: $\lambda$, $\rho$, T, t, RC | | |
|---|---|---|---|---|
| | | $R^2$ | RMSE | MAE |
| $\eta > 6000$ cP | Train | 0.89 | 443 | 359 |
| | Test | 0.88 | 543 | 475 |
| $\eta < 6000$ cP | Train | 0.95 | 641 | 531 |
| | Test | 0.88 | 1071 | 903 |
| | | Set 2: SA, $\rho$, T, t, RC | | |
| | | $R^2$ | RMSE | MAE |
| $\eta > 6000$ cP | Train | 0.89 | 437 | 353 |
| | Test | 0.88 | 534 | 458 |
| $\eta < 6000$ cP | Not found | | | |
| | | Set 3: k, $\rho$, T, t, RC | | |
| | | $R^2$ | RMSE | MAE |
| $\eta > 6000$ cP | Train | 0.89 | 434 | 328 |
| | Test | 0.88 | 486 | 391 |
| $\eta < 6000$ cP | Train | 0.95 | 678 | 582 |
| | Test | 0.88 | 1118 | 990 |
| | | Set 4: k, T, t, RC | | |
| | | $R^2$ | RMSE | MAE |
| $\eta > 6000$ cP | Train | 0.86 | 490 | 386 |
| | Test | 0.85 | 604 | 518 |
| $\eta < 6000$ cP | Train | 0.94 | 749 | 327 |
| | Test | 0.89 | 1040 | 917 |

Gray highlights represent the selected models.

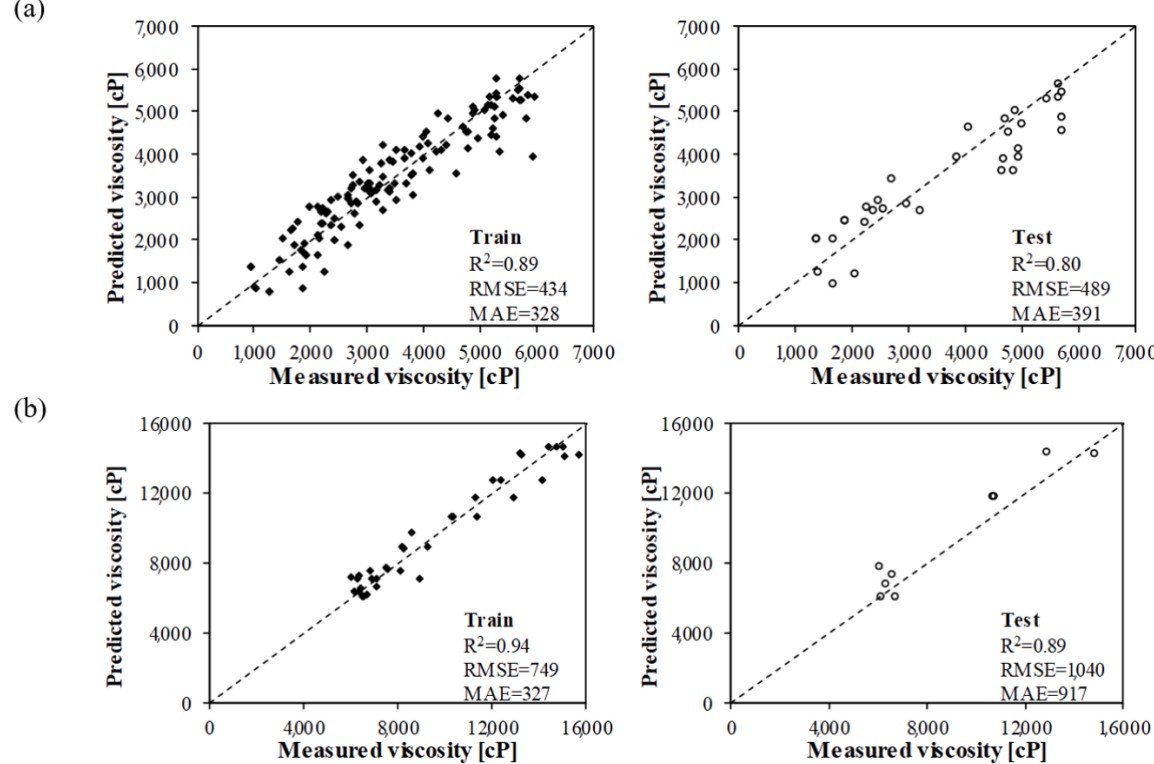

**Figure 5.** Predictive models for (**a**) $\eta < 6000$ cP—Inputs: k, $\rho$, T, t, RC, and (**b**) $\eta > 6000$ cP—Inputs: k, T, t, RC.

Equation (8) was used for predicting two sets of data that were not included in the analysis described above. This was intended to be an independent validation of the model. Figure 6 shows a comparison of the measured and predicted viscosities using the final MGGP models.

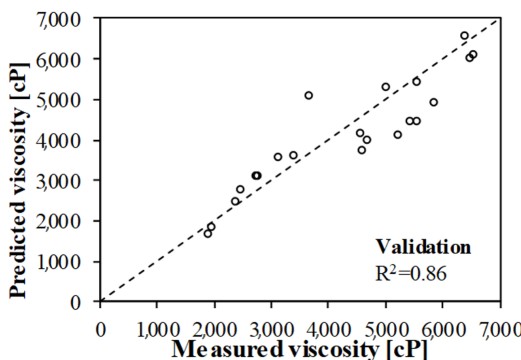

**Figure 6.** Results of the validation process.

## 5. Conclusions

The magnitude of the interaction mechanisms between CR and bitumen during the different stages of the blending process is highly variable. The ability to forecast the effect of the mixing process for a given combination of binder and crumb rubber and a given set of processing conditions may be of great interest to practitioners and promote the use of rubber from ELTs for paving applications. For this purpose, the Multi-Gene Genetic Programming algorithm was used to develop a predictive model of CR-modified binders' viscosities using the physical properties of the crumb rubber and the mixing process characteristics as the inputs.

The results obtained by the application of the MGGP algorithm on the whole database, as well as those obtained using the Multi-Variable Least Square Regression (MLSR) method, showed that the overall goodness of fit of the MGGP model ($R^2 = 0.89$) was excellent compared to the model obtained through the MLSR approach. In particular, the MLSR appeared to be inadequate due to its mathematical form. A detailed analysis of the results showed a significant difference in predicting viscosity values below ($R^2 = 0.72$) and above ($R^2 = 0.81$) 6000 cP, which were mainly collected on binders modified with a high CR content. This suggested that certain characteristics of both the crumb rubber particles and mixing process, may have a specific significance and impact during the different blending phases. Based on these findings, the MGGP model was refined in order to have separate models for viscosity above and below the threshold value of 6000 cP. A sensible improvement of the goodness of fit was obtained and the comparison of the measured and predicted viscosities of the dataset used for the validation process showed excellent performance. The application of the Multi-Gene Genetic Programming algorithm is very promising since it overcame the problems encountered when using the MLSR approach. Moreover, unlike other machine learning algorithms, the MGGP returned an actual equation, which can be used by practitioners to forecast the interaction effects in terms of viscosity and tune the production process accordingly.

**Funding:** This research received no external funding.

**Institutional Review Board Statement:** Not applicable.

**Informed Consent Statement:** Not applicable.

**Data Availability Statement:** All data, models, or code that support the findings of this study are available from the corresponding author upon request.

**Conflicts of Interest:** The author declares no conflict of interest.

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
