# Peer review of "Soft Computing Approach for Predicting the Effects of Waste Rubber–Bitumen Interaction Phenomena on the Viscosity of Rubberized Bitumen"

_sustainability, doi:10.3390/su142113798_

Round 1

Reviewer 1 Report

The effect of rubber-bitumen interaction on the viscosity of rubberized bitumen considering various factors such as particle size, rubber content, temperature, etc. was experimentally investigated, and two algorithms of MLSR and MGGP were used to predict the effects of rubber-bitumen interaction phenomena on the viscosity of rubberized bitumen. The topic is benefit for the Sustainability of pavement materials, the purpose of this research is clearly stated and the data provided are sufficient. 

(1)    The authors stated that MLSR model was compared to a preliminary MGGP-based model to HIGHLIGHT THE BENEFITS of using the MGGP algorithm, how can MLSR bring out the advantage of MGGP if the former algorithm is not suitable for prediction? Please justify and clarify in detail.

(2)    The model only was verified by the data from the presented research, the reviewer suggests the authors use other data from literatures to expand the applicability of the model.

Author Response

Please see file attached.

Reviewer 2 Report

General Comments

The submitted paper sustainability-1881941 analysis the use of the Multi-Gene Genetic programming tool to predict viscosity changes, due to interaction phenomena, in rubberised asphalt binders. The manuscript is well structured and agrees with the journal’s aim and scope.

The paper’s topic is well described, particularly the development of the predictive model gives the possibility to further formulations by other authors. Materials and Methods are well explained and motivated, and the measurement steps are detailed. Figures agree to journal specifications except for Figure 3, its DPI must be improved.

Author Response

Please see file attached.

Reviewer 3 Report

Reviewer’s comment:

This manuscript deals with the interaction between waste rubber particles from end-of-life tires and bitumen. In all, the paper falls within the scope of this journal and is written in a clear way and most of the claims are supported by data and figures. It is of great significance to scholars and the results are very important significance for industry manufacturers. The manuscript is relatively well organized, and can be accepted after a minor revise.

Figure 2 shows the relationship between the predicted and measured viscosity values using the MLSR and MGGP approaches. MGGP model shows a good overall goodness-of-fit, and a difference exists for viscosity values above 6000cP in MLSR model. Can the author consider that 6000cP is the threshold value of rubber particles content in bitumen?

Is there any proof that can prove the threshold value of 6000cP, such as mechanical properties or thermal properties? Please comment it.

Author Response

Please see file attached.
